# Tracing back variations in archaeal ESCRT-based cell division to protein domain architectures

**Béla P. Frohn**[1,2], **Tobias Härtel**[1], **Jürgen Cox**[2,3]*, **Petra Schwille**[1]*

**1** Department of Cellular and Molecular Biophysics, Max-Planck Institute of Biochemistry, Martinsried, Germany, **2** Computational Systems Biochemistry Research Group, Max-Planck Institute of Biochemistry, Martinsried, Germany, **3** Department of Biological and Medical Psychology, University of Bergen, Bergen, Norway

* cox@biochem.mpg.de (JC); schwille@biochem.mpg.de (PS)

**Data Availability Statement:** All relevant data are within the paper and its Supporting Information files.

**Funding:** The author(s) received no specific funding for this work.

## Abstract

The Endosomal Sorting Complex Required for Transport (ESCRT) system is a multi-protein machinery that is involved in cell division of both Eukaryotes and Archaea. This spread across domains of life suggests that a precursor ESCRT machinery existed already at an evolutionary early stage of life, making it a promising candidate for the (re)construction of a minimal cell division machinery. There are, however, only few experimental data about ESCRT machineries in Archaea, due to high technical challenges in cultivation and microscopy. Here, we analyse the proteins of ESCRT machineries in archaea bioinformatically on a protein domain level, to enable mechanistical comparison without such challenging experiments. First, we infer that there are at least three different cell division mechanisms utilizing ESCRT proteins in archaea, probably similar in their constriction mechanisms but different in membrane tethering. Second, we show that ESCRT proteins in the archaeal super-phylum Asgard are highly similar to eukaryotic ESCRT proteins, strengthening the recently developed idea that all Eukaryotes descended from archaea. Third, we reconstruct a plausible evolutionary development of ESCRT machineries and suggest that a simple ESCRT-based constriction machinery existed in the last archaeal common ancestor. These findings not only give very interesting insights into the likely evolution of cell division in Archaea and Eukaryotes, but also offer new research avenues by suggesting hypothesis-driven experiments for both, cell biology and bottom-up synthetic biology.

## Introduction

Cell division is one of the most fundamental biological processes and typically regarded as one of the basic properties of life. It must be executed in a highly controlled manner, as errors can quickly have fatal consequences. The emergence of cell division machineries that allowed such a high degree of control was hence a major milestone in early evolution, enabling cells to regulate their reproduction. However, how such an early cell division machinery might have looked like is not yet known, and it has not yet been possible to build an *in vitro* system that can

**Competing interests:** The authors have declared that no competing interests exist.

mimic early cell division by fully dividing a membrane vesicle. Thus, the important question of how cell division emerged as a mechanism remains unsolved.

One membrane-transforming machinery which is particularly interesting to this open question is the Endosomal Sorting Complex Required for Transport (ESCRT) [1–4]. It is involved in cell division of both Eukaryotes and Archaea, so it is likely that ESCRT proteins were already playing a role in cell division in their last common ancestor. Especially, it has been suggested that the mechanistic principles of ESCRT machineries in Eukaryotes and Archaea are comparable [1,2,5,6], indicating that ESCRT-like proteins in the common ancestor already formed a machinery. Comparative studies of the variations of today's known ESCRT machineries hence promise to give insights into an evolutionary early mechanism of cell division.

While the eukaryotic ESCRT machinery has become a highly popular research target over the last years, as it is potentially involved in a wide range of human diseases [7–9], progress in researching the archaeal pendant is slow. One reason for this is that most archaea thrive only under extreme conditions, what makes microscopic or molecular biology-based studies technically challenging. This lack of information is particularly unfortunate when one is interested in a potential ancient ESCRT machinery, because a comparative approach investigating similarities between machineries derived from the same ancestor offers more significant insights when including an higher diversity of descendants in the analysis. Thus, a deeper understanding of the mechanisms of archaeal ESCRT machineries and their diversity is needed.

The ESCRT machinery in archaea was first found in *Sulfolobus acidocaldarius* [3,4], and it was originally named Cdv system (Cdv for Cell DiVision). In *S. acidocaldarius*, the machinery consists of three proteins, named CdvA, CdvB (having multiple paralogs) and CdvC [1–4]. Importantly, CdvB proteins are homologs of the eukaryotic ESCRT-III proteins (also having multiple paralogs), and CdvC is a homolog of the eukaryotic Vps4 protein, which is also a part of the ESCRT machinery [1–4]. CdvA, however, has no homologs in Eukaryotes. Thus, there are both similarities and differences in the composition of the machineries, supporting the hypothesis that they derived from a common ancestral machinery but evolved differently.

After the discovery of CdvABC in *S. acidocaldarius*, homologous proteins were identified in many other archaea. There are four archaeal super-phyla, named DPANN, Euryarchaeota, Asgard and TACK [10], and three of them contain species that possess Cdv proteins [11]. In Euryarchaeota, some few species show CdvB and/or CdvC proteins encoded in their genome, but most species do not possess Cdv proteins. In Asgard archaea, only little is known, because it was just recently possible to cultivate a member of this super-phylum [5,12], but metagenome data and protein prediction indicates that genes encoding for CdvB and CdvC proteins are common within this group, and first experimental evidence is accumulating [13,14]. In TACK archaea, to which *Sulfolobus acidocaldarius* belongs, species of all groups except Thermoproteales possess all three proteins CdvABC. These variations between super-phyla once more indicate that there was once an ancient Cdv/ESCRT machinery that has evolved differently. Importantly, it was recently suggested that all Eukaryotes originate from the Asgard super-phylum [15], so the eukaryotic ESCRT machinery might be a highly refined version of the Asgard Cdv system. This idea emphasizes the relevance of studying archaeal Cdv machineries to increase our understanding of ancestral versions of eukaryotic ESCRT machineries.

Surprisingly, experiments conducted with TACK archaea showed unequal results in the two main groups of this super-phylum, Crenarchaeota and Thaumarchaeota, although in both groups all three proteins CdvABC are found [1,2]. While in both groups fluorescence microscopy studies showed that initially CdvA proteins enrich at the future division site, probably binding to the membrane, the subsequent steps seem to differ. In Crenarchaeota, CdvB homologs successively join CdvA, together with CdvC [1,5,6,16–19]. While the exact mechanism is

unknown, it is mostly suggested that CdvB homologs form a ring-like higher order structure, tethered to the membrane by CdvA [1,16,18]. Then, it is assumed that this ring constricts with the help of CdvC, leading to membrane fission and cell-division [1,5,6,17]. In contrast, in the Thaumarchaeon *Nitrosopumilus maritimus*, only CdvC concentrates at the division site together with CdvA, while the distribution of CdvB is diffuse [1,20,21]. Up to now, there exists no explanation for this deviation and no model of how membrane constriction may be achieved in Thaumarchaeota.

Thus, while a comparative analysis of Cdv/ESCRT machineries promises interesting insights into an evolutionary early mechanism of cell division, there are two major challenges. First, experimental studies with archaeal Cdv machineries are technically highly challenging, due to the extreme conditions archaeal organisms thrive under. Second, comparing presence or absence of homologs between organisms seems insufficient to draw conclusions about differences in the emerging functions: Both Cren- and Thaumarchaeota possess proteins classified as CdvABC, but nonetheless there seem to be differences in their machineries. Hence, simply investigating which proteins of the Cdv/ESCRT machinery might have existed in the ancient predecessor, as done in previous studies [1,2], cannot lead to a deep mechanistical understanding.

Thus, we here analyse the proteins of archaeal Cdv machineries bioinformatically on a domain level, going beyond the pure existence or absence analysis. This is inspired by the idea that the mechanism of a protein machinery is a result of the interactions between the proteins, and that these interactions occur at specific regions of the proteins' amino acid sequences. We show that CdvABC homologs in Cren- and Thaumarchaeota differ in their domain compositions and that the different interactions of the proteins caused by these different compositions might explain the previously observed experimental variations. Backed by this finding we then expand the domain analysis to Asgard archaea and find that Cdv machineries in this archaeal super-phylum are highly similar to the eukaryotic ESCRT machinery. Finally, by combining the domain analysis with phylogenetic analysis, we propose an evolutionary scenario of Cdv/ESCRT development, which provides insights into how an ancient machinery based on evolutionary early Cdv/ESCRT proteins might have looked like. Importantly, we suggest that this ancient machinery might have been able to form a contracting ring, making it a strong candidate for a very simple cell division machinery.

## Results

### Cdv systems are composed of eleven domains, seven of them occurring in all organisms, four only in specific phylogenetic groups

Based on homology to previously studied Cdv machineries, we searched publicly available genomes and proteomes of archaeal organisms for homologs of CdvABC proteins. From the resulting list of organisms containing at least one CdvABC homolog we selected 37 organisms based on the high quality of their genomic information. We then identified domains of interest in Cdv proteins of the selected organisms by secondary structure prediction, by search against Pfam, InterPro and the Conserved Domain Database (CDD), and by utilizing regular expressions reported in the literature. Hereafter we will call all regions in the amino acid sequences of proteins that are expected to execute a distinct function *domains*, although strictly speaking some are only motifs). As **Fig 1** shows, we found eleven different domains in Cdv machineries (for the exact locations see **S1 Table**). Four domains (Snf7, AAA+ATPase, MIT and Vps4_C) were found in organisms of all three super-phyla, indicating that they were already part of evolutionary early Cdv machineries. One domain (MIM2) was found in Asgard and TACK archaea, but not in Euryarchaeota, two domains (CdvA_alpha and CdvA_beta) were found in

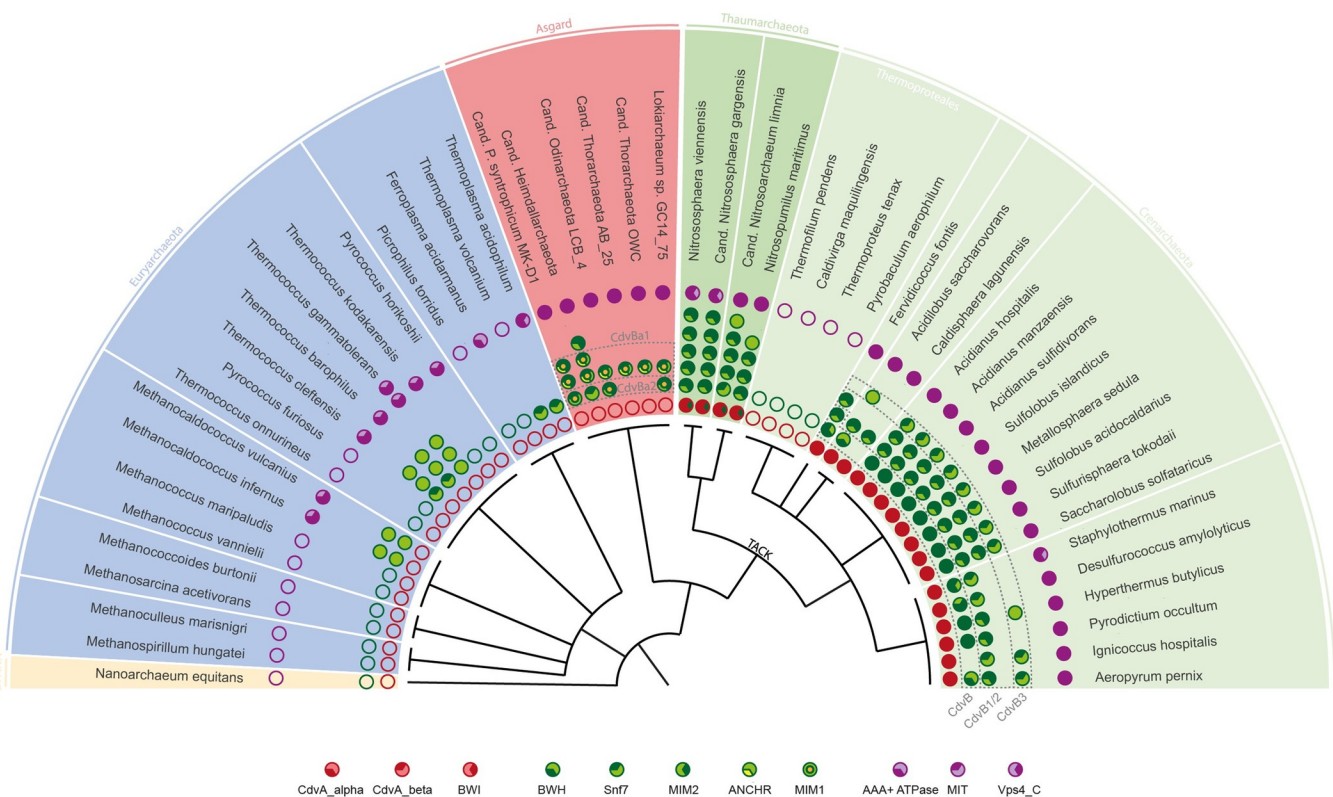

**Fig 1. Distribution of CdvABC homologs in archaea and their domain composition.** Existing homologs are displayed as filled circles, missing homologs as empty circles. Red circles represent CdvA homologs, green circles represent CdvB homologs and purple circles represent CdvC homologs. Where parts of the sequence of a homolog could be assigned to specific protein domains, this is visualised by segmentation of the filled circles with deeper colour. Organisms are arranged by phylogenetic relationship [25] and coloured by super-phyla. The TACK super-phylum is further divided into Thaumarchaeota and Crenarchaeota. 51 organisms are display, 37 of them possess at least one CdvABC homolog, 14 do not possess any CdvABC homologs. Abbreviations: BWI: Broken Winged-Helix Interaction Site, CdvA_alpha: Alpha-helix rich CdvA domain, CdvA_beta: Beta-sheet rich CdvA domain, BWH: Broken Winged Helix domain, MIM1 / -2: MIT-interacting-motif 1 / -2, Snf7: Vacuolar-sorting protein SNF7 domain, Vps4_C: Vps4 C terminal oligomerisation domain, MIT: Microtubule Interacting and Trafficking molecule domain, ANCHR: N-terminal membrane binding domain in ESCRT-III proteins.

TACK archaea, but not in Euryarchaeota or Asgard archaea, and two domains (BWI and BWH) were found only in the TACK subgroup called Crenarchaeota. This suggests that by gain or loss of some domains, rather than gain or loss of full proteins, Cdv systems developed differently from their common ancestor, providing a hint that this might be an explanation for differences in their respective mechanisms. Finally, in Asgard archaea, we found two more unique domains. Interestingly, these matched domains occurring in the eukaryotic ESCRT machinery. First, consistent with the results of Lu et al. [22], we found potential MIM1 [23] domains at the C-terminus of most Asgard CdvB homologs. Second, consistent with the results of Caspi and Dekker [1], we predicted alpha-helices at the N-terminus of one group of Asgard CdvB homologs that match the ESCRT-III N-terminal ANCHR [24] motif. Thus, this simple analysis already yielded two interesting results: First, four domains are shared between all organisms included in this study, which suggests that the Cdv present in their common ancestor likely presented these domains. Second, some domains occur only in specific phylo-genetic groups, thus they are promising candidates to explain differences between the machin-eries of these groups. To analyse these differences and the machineries they might give rise to, we next investigated the domain architectures of proteins within each super-phylum in detail, starting with Euryarchaeota.

## Cdv protein domains in Euryarchaeota do not constitute a common functional system

Only few Euryarchaeota showed CdvABC encoding genes, and in half of these no conserved domain sequences could be found (blue background in **Fig 1**). This indicates that Cdv protein genes in Euryarchaeota mutated beyond domain-recognition. Furthermore, no domains allowing interaction between proteins could be identified, making a cell-division system similar to Cdv-including mechanisms in other archaea unlikely. Instead, most of them possess multiple FtsZ homologs (**S2 Table**), which at least in some species were experimentally shown to be crucial for cell division [26,27], and it is widely accepted that Euryarchaeota divide by a FtsZ-based mechanisms more comparable to bacteria than other archaea [1]. Thus, our findings match to this idea that Euryarchaeota do not use a Cdv-including system for cell division. However, these proteins might still play physiological roles in machineries different from Cdv/ ESCRT in TACK, Asgard and Eukaryotes.

## Cdv proteins of Cren- and Thaumarchaeota are composed of different domains

In contrast to Euryarchaeota, in TACK archaea we identified CdvABC homologs in all phylogenetic orders except Thermoproteales, where it was suggested that cell division is based on the actin homolog crenactin [28], although this idea has recently been questioned [1]. Furthermore, in nearly all Cdv proteins of TACK archaea we identified specific conserved domains in the Cdv proteins, allowing comparison of domain architectures between organisms (**Fig 1**, green background). This comparison showed architectures that were relatively consistent within phylogenetic orders, meaning that within organisms of the same phylogenetic order homologous proteins mainly consist of the same domains. This suggests that mechanisms may be comparable between organisms of the same phylogenetic order. On the level of phyla, however, domain architecture showed distinct differences: While in all Thaumarchaeota the proteins classified as CdvA homologs were constructed of the three domains CdvA_alpha, CdvA_beta and MIM2 (or putative MIM2), in all organisms of Crenarchaeota the proteins were composed of CdvA_alpha, CdvA_beta and BWI. Furthermore, while the great majority of Crenarchaeota possessed exactly one protein with a BWH domain, classified as CdvB homolog, this domain did not occur in Thaumarchaeota at all. Importantly, BWI and BWH are known to be able to interact [16,18,19], so the joint absence or presence of the two domains indicated a mechanistical difference in the machineries of Cren- and Thaumarchaeota. To further scrutinize this finding, we next analysed which protein-protein-interactions (PPIs) may be possible based on the domain architecture in TACK archaea.

## Protein domain compositions explain experimental findings indicating differences between Cren- and Thaumarchaeota

To infer PPIs, we searched in the literature which interactions the domains we had identified might allow (**Figs 2 and S1**). We found that four domain-pairs are experimentally shown to lead to interaction: Two of these interactions are between different domains, that is MIM2-MIT [1,18,29] and BWI-BWH [16,18,19], and two of them are between the same domains, that is Snf7-Snf7 [1,19] and Vps4_C-Vps4_C [1,22]. Furthermore, it was suggested that CdvA_beta can polymerise [18,19], but experiments on this are not conclusive [1]. Importantly, these experiments were conducted with only a selection of the organisms included in this study. Our approach here was to test whether generalisation of these interactions can lead

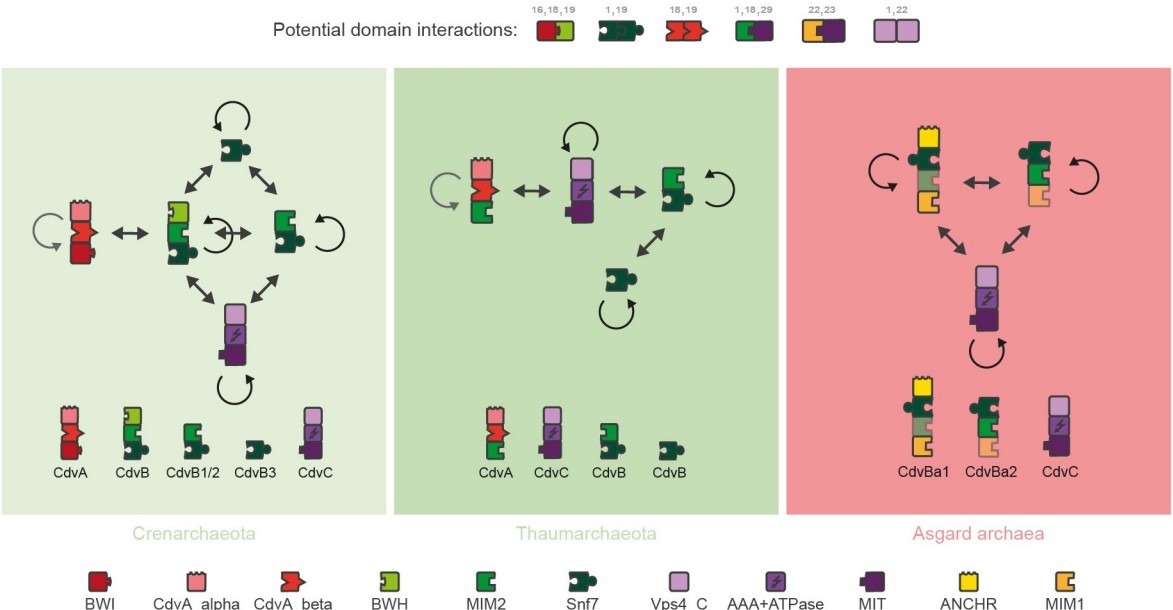

**Fig 2. Potential protein-protein interactions of archaeal CdvABC homologs.** Proteins are displayed as sets of building blocks, each building block representing one protein domain. Five potential domain-based interactions derived from literature are displayed in the top, grey numbers indicating the literature references where these domain-domain interactions were described. Based on the potential domain-domain interactions, protein-protein interaction networks are shown, where arrows indicate a potential interaction between two domains of the connected proteins. Literature about polymerisation of the CdvA_beta domain varies, so it is displayed in grey instead of black. The networks visualise the interactions of proteins that most likely existed in the last common ancestors of each of the three phylogenetic groups Crenarchaeota, Thaumarchaeota and Asgard archaea (see also **Fig 6**). Thus, this visualises conceptual differences between the three phylogenetic groups, and does not reflect the specific protein interactions of all organisms found today. Abbreviations and colours as in **Fig 1**.

to interesting results. Thus, we asked which interaction networks would arise if interactions were governed by domains and the interaction rules just described.

In Thaumarchaeota, our approach suggested four qualitatively different potential PPIs. First, CdvA might interact with CdvC based on the MIM2 domain in CdvA and the MIT domain in CdvC. Second, the same interaction might take place between most of the CdvB homologs and CdvC. Third, most CdvB homologs might be able to polymerise at the Snf7 domain. Fourth, CdvC proteins might polymerise as well, utilizing the Vps4_C domain. In contrast, in Crenarchaeota our approach also suggested four possible PPIs, but these were based on other domains and occurred between different proteins. First, the BWI domain of CdvA might bind to the one CdvB homolog possessing a BWH domain. Second, all CdvB homologs might bind to each other and polymerise utilizing the Snf7 domain. Third, the CdvB homologs which possess a MIM2 domain might be able to interact with the MIT domain of CdvC. Fourth, CdvC might polymerise at the Vps4_C domain. Thus, there is one major difference from the suggested PPIs of Thaumarchaeota to the ones of Crenarchaeota (**Fig 2**): while in Thaumarchaeota the domain composition suggests that CdvA binds to CdvC, in Crenarchaeota it implies that CdvA binds to CdvB. Interestingly, this finding matches well to the differences between the two phyla found in fluorescence microscopy studies [1,5,6,16,19–21]. In the Thaumarchaeon *N. maritimus*, fluorescence bands of CdvA at the division site are followed by bands of CdvC, whilst in Crenarchaeota CdvA bands are followed by CdvB bands.

Thus, these previously unexplained experimental differences support our analysis of potential PPIs. Furthermore, in Crenarchaeota it was experimentally shown that CdvB homologs enrich sequentially at the division site, first the CdvB homolog possessing a BWH domain,

second the CdvB homologs not possessing this domain [5]. Again, this is what the domain-architecture-based PPI inference suggests, as the enrichment is thought to be mediated by CdvA, and in our suggested PPI network only the CdvB homolog with a BWH domain can interact with CdvA. Reassured by this support from the literature, we next tried to derive mechanistical models based on the domain compositions for membrane constriction and cell division in TACK archaea, beginning with Crenarchaeota.

## Mechanistical models of Cdv based cell division suggest two different systems in TACK archaea

Crenarchaeota organisms showed different numbers of CdvB homologs (**Fig 1**). As we wanted to derive a common mechanism for Crenarchaeota, which might be slightly adapted by individual organisms, we first had to make homologs comparable to find similarities within the phylum. Thus, we clustered the homologs by phylogeny (**Fig 3**) and then analysed the domain architectures within the clusters. Reassuringly, the differences in domain architecture were congruent with their phylogeny, as seen by their placement in different branches within the CdvB tree (**Fig 3**). Additional gene cluster analysis showed similar architectures of Cdv genes within phylogenetic branches (**S2 Fig**). Therefore, we defined three different classes of CdvB homologs, which we named CdvB, CdvB1/2 and CdvB3 with respect to previous work [1]. CdvB class proteins are constructed of the three domains Snf7, MIM2 and BWH, CdvB1/2 class proteins of Snf7 and MIM2, and CdvB3 class proteins of Snf7 only (**Fig 1**). Because this result implies that the common ancestor of all Crenarchaeota did possess these three CdvB homologs, we based our mechanistical model on these proteins (together with CdvA and CdvC). Then, to infer a mechanistical model, we first assigned one specific function to each domain (**Fig 4** legend, white background), derived from literature, and defined one protein as the initiating one. Second, we analysed possible PPIs step-by-step and thereby deduced possible occurring higher order processes (**Figs 4 and S1**). Importantly, this is not a mathematical modelling, but a qualitative explanation.

In Crenarchaeota (**Fig 4A**), we defined CdvA as initiating protein, because it is the only protein with a potentially membrane-targeting domain, and experiments show that neither CdvB nor CdvC can bind the membrane [1,16,18,19]. Also, it has been validated that CdvA is the first protein to enrich at the division site [1,16,18,19]. As we had identified only one PPI for CdvA in Crenarchaeota, BWI-BWH, we first suggest that the CdvB class homolog binds to CdvA. This protein then has two more potential PPI options, mediated by the Snf7 and MIM2 domains. Which of these options is utilized first cannot be inferred from our analyses, so we had to consult the literature. It was shown experimentally that CdvB, CdvB1/2 and probably also CdvB3 enrich at the division site before CdvC joins [1,6]. Thus, we suggest that the affinity for the Snf7-Snf7 interaction is higher than for the MIM2-MIT interactions, and hence, that a ring-like structure composed of CdvB, CdvB1/2 and probably also CdvB3 is formed. Thus, while we cannot directly infer from our analyses why CdvB polymerisation takes place before CdvC enrichment, we can explain why CdvB enriches before CdvB1/2 and CdvB3. We strongly encourage experimentalist to compare Snf7-Snf7 and MIM2-MIT affinities to test this idea.

After the ring has been assembled at the division site, composed of CdvB homologs and tethered to the membrane by CdvA, there are two domains left to interfere with: the MIM2 domain of the CdvB class homolog and the MIM2 domain of the CdvB1/2 class homolog. As experimental data suggests that CdvB class homologs enrich before CdvB1/2 class homologs [5], we asked which structural cause this could have. The interaction is reported to take place at the short motif MIM2 [1,29]. Importantly, in the MIM2 motif of ESCRT-III there are four

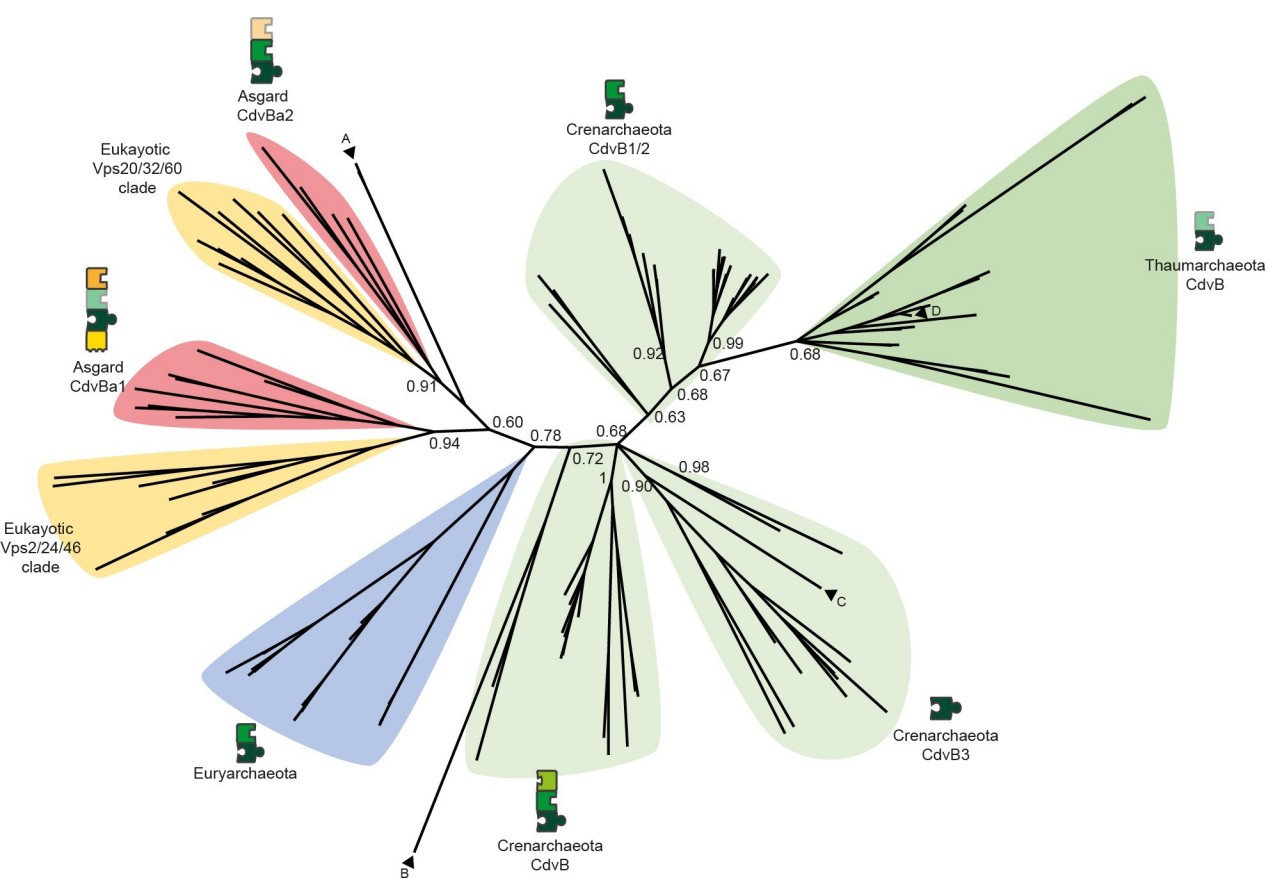

**Fig 3. Phylogenetic tree of CdvB homologs calculated via Bayesian phylogeny.** Background colours as in Fig 1 plus Eukaryotes (ESCRT-III proteins of *Homo sapiens* and *Saccharomyces cerevisiae*) in yellow, proteins schematically depicted as sets of domains as in Fig 2. In Crenarchaeota, CdvB homologs split into three phylogenetic clusters (CdvB, CdvB1/2 and CdvB3) where within each cluster most homologs have the same domain composition. In Thaumarchaeota, no clear phylogenetic clustering is visible. In Asgard archaea, CdvB homologs split into two branches, with clear differences in domain architecture between clusters. They group together with the two eukaryotic ESCRT-III groups Vps2/24/46 (CdvBa1) and Vps20/32/60 (CdvBa2), indicating a shared evolutionary history of ESCRT-III proteins in Asgard archaea and Eukaryotes. Single proteins that do not match the clusters are indicated by arrows. These might be the results of horizontal gene transfer or contamination of metagenome data. A: Two Euryarchaeota CdvB proteins (*Thermoplasa acidophilum*, UniProt ID Q9HIZ5 and *Thermoplasma volcanium*, Q97BR8). B: *Thaumarchaeon Nitrosopumilus maritimus*, A9A4K8. C: *Fervidococcus fontis*, domain architecture of a CdvB protein (Snf7 and MIM2), I0A2N3. D: *Candidatus Heimdallarchaeon*, A0A523XLA6.

prolines which seem to be essential [29]. Interestingly, in the CdvB1/2 class homologs we found consistently fewer (2 instead of 4) prolines than in the CdvB class homologs (**S3 Fig**). Thus, it is tempting to speculate that this lower number of prolines might cause a lower affinity of the MIM2 domain in CdvB1/2 class homologs to the MIT domain in CdvC than the MIM2 domain in CdvB class homologs. This higher affinity of CdvB class homologs would be a simple mechanistical explanation of the experimental results. Giving respect to the literature, in our hypothesised mechanism we suggest that in the next step CdvC binds preferably to the CdvB class homologs in the ring at the division site.

Based on the AAA+ ATPase and Vps4_C domains of CdvC, this binding presumably results in the disassembly of the bound CdvB class homolog from the higher-order structure [1]. As more and more CdvB class homologs get disassembled, CdvC more often binds to CdvB1/2 class proteins, due to the increasing concentration, and starts disassembling them, too. Thus, the ring of homologs first becomes a ring of mostly CdvB1/2 and CdvB3 class homologs, and then gets disassembled. Once again, this is in accordance with experimental

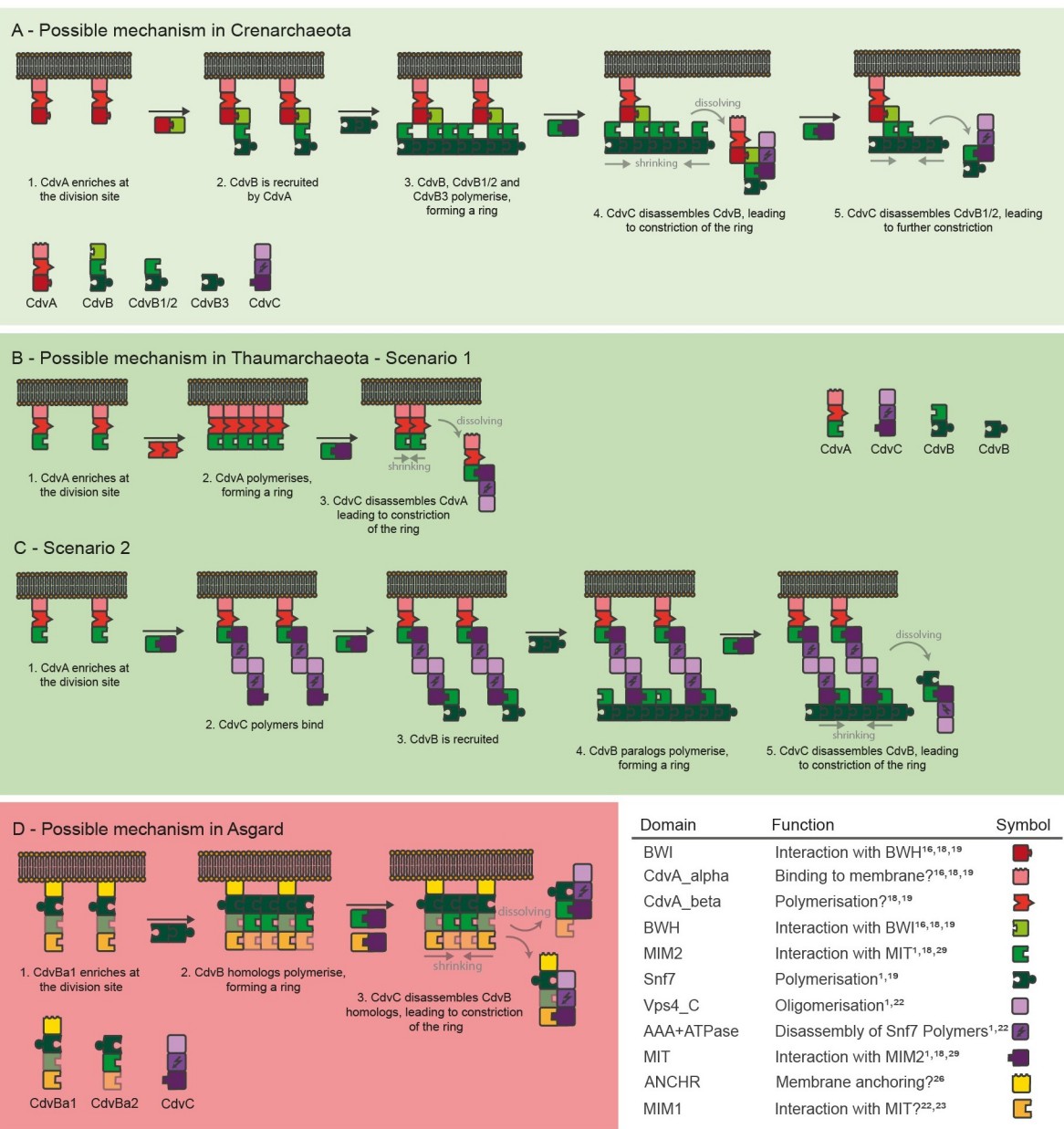

**Fig 4. Possible mechanisms of Cdv-based cell division in Crenarchaeota, Thaumarchaeota and Asgard.** Proteins are displayed as sets of building blocks as in Fig 2. Importantly, these mechanisms are not results of mathematical modelling, but are a qualitative explanation based on PPI networks (**S1 Fig**). **A**: The possible mechanism in Crenarchaeota is characterised by a sequential enrichment of CdvA, CdvB, CdvB1/2 and -3, followed by depolymerisation by CdvC. **B**: In the first scenario in Thaumarchaeota, CdvB homologs are not involved at all and ring-formation is instead relying on polymerisation of CdvA. **C**: The second scenario in Thaumarchaeota involves CdvB polymerisation, which is connected to CdvA by CdvC acting as linker. We strongly doubt this scenario. **D**: In Asgard archaea, the possible mechanism starts by CdvBa1 homologs binding to the membrane, differing from TACK archaea. CdvB homologs may then polymerise, form a ring, and get depolymerised by CdvC.

data [5]. However, we cannot explain how CdvB3 class homologs can dissociate from the division site, as we did not identify domains mediating interaction with CdvC in these homologs. Finally, how this mechanism results in membrane constriction and cell division can only be speculated, and we will suggest possible explanations in the Discussion.

In Thaumarchaeota, a classification of CdvB homologs as in Crenarchaeota was not possible. Phylogenetic and gene cluster analyses showed no clear clusters (**Figs 2 and S2**), and the domain composition of most homologs was the same: Snf7 and MIM2 (**Fig 1**). Thus, the role of different CdvB homologs in Thaumarchaeota remains unclear. Nonetheless, we were able to propose a possible mechanism based on the domain compositions (**Figs 4B, 4C and S1**).

Same as in Crenarchaeota, the suggested process starts with CdvA binding to the membrane and enriching at the division site. Next, CdvC binds to CdvA, utilizing the MIM2-MIT interaction. Then, there are two options: either CdvA itself had already formed a ring-like polymer at the division site, which is then disassembled in the same way as CdvB in Crenarchaeota (**Fig 4B**). Alternatively, CdvC functions as a linker to CdvB, which could then result in a mechanism similar to Crenarchaeota (**Fig 4C**). However, in the second scenario it is unclear how the higher-order structure could eventually be disassembled again, as CdvC can hardly function as a linker between CdvA and CdvB, and as a disassembler of CdvB at the same time. Therefore, we strongly prefer the first scenario, which is also supported by the diffuse distribution of CdvB in fluorescence microscopy experiments [20]. Driven by the good agreement of our inferred models with experimental data, we next used the same approach to investigate Asgard archaea.

## Asgard archaea possess a system closely related to ESCRT

All investigated genomes of Asgard archaea contained at least one CdvB and one CdvC homolog, indicating their conservation within the super-phylum (**Fig 1**). Furthermore, in the genome of the only Asgard archaeon that could yet be cultivated and whose genome data is therefore trustworthy (*Candidatus Prometheoarchaeum syntrophiculm*) [12], all CdvB homologs contained either MIM1 or MIM2 domains, both of which could allow interactions with the MIT domain found in the CdvC homolog. Potential interactions between CdvBs and CdvC based on these domains in Asgard archaea were recently investigated in detail by Lu et al. [22], whose results suggest that indeed CdvC in Asgard archaea can disassemble CdvB filaments, supporting the idea of a Cdv-including mechanism comparable to other archaea in this super-phylum. Another recent experimental study further supports the idea of Cdv protein-protein interactions in Asgard archaea similar to Crenarchaeota and Eukaryotes [13].

However, to form a cell division system, this potential constriction machinery must be tethered to the membrane. But there is no CdvA homolog in Asgard archaea to achieve this, so we had to search for different ways of membrane tethering. As in our model every function of a protein is based on domains, for membrane binding there were two options: either some of the CdvB homologs in Asgard archaea possess domains that can bind to the membrane themselves (similar to ESCRT-III), or they possess a domain that can interact with another protein executing this task (similar to Crenarchaeota). Caspi and Dekker [1], who at that time had only metagenome data of a single Asgard archaeon (*Lokiarchaeon sp. GC14_75*) available, hypothesised that the N-terminal alpha helix similar to the ESCRT-III ANCHR motif [24] found in one Lokiarchaeon CdvB homolog might allow direct interaction with the membrane. Our analysis of multiple Asgard archaea genomes and especially that of its only cultivated member *Cand. P. synthrophicum* indeed revealed that this region is highly conserved within Asgard archaea, but only in the CdvBa1 subgroup of Asgard CdvB homologs (**Fig 5A**). Interestingly, in our phylogenetic tree this subgroup clusters with the eukaryotic Vps2/24/46 group of ESCRT-III proteins (**Fig 2**), which are the ESCRT-III versions that can bind to the membrane via the ANCHR motif [24]. Thus, it is plausible to assume that the membrane binding of the Vps2/24/46 ESCRT-III group is similar to the CdvBa1 group of Asgard archaea. Supporting this idea, secondary structure prediction suggested that the region forms an alpha helix,

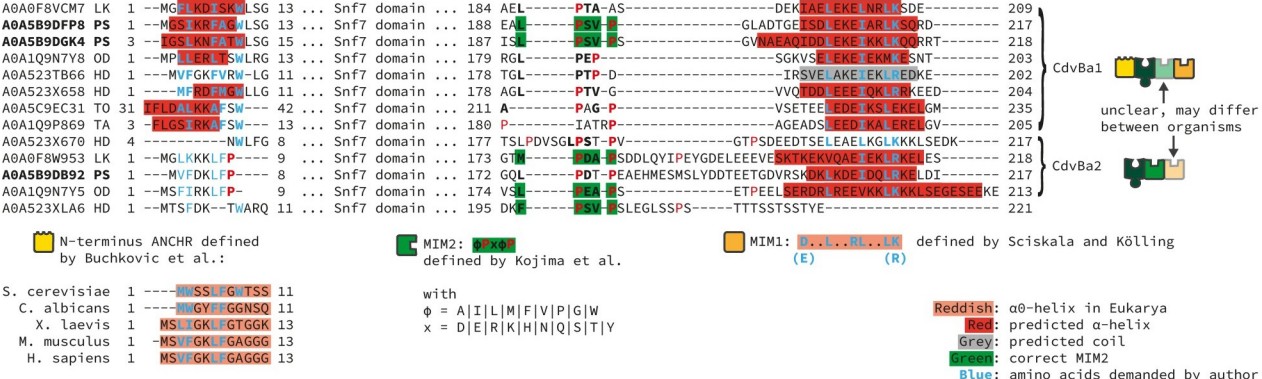

**Fig 5. Multiple sequence alignment and potential domains of Asgard CdvB homologs.** Predicted secondary structure is indicated by background colour, amino acids matching regular expressions of specific domains as defined in the literature are indicated by font colour. Prolines in red. Protein sequences of the different clades in the phylogeny (**Fig 3**), CdvBa1 and CdvBa2, differ mostly at the N-terminus, where in all but one sequences of CdvBa1 an alpha helix is predicted, whilst in in all CdvBa2 proteins no alpha helix is predicted. Abbreviations: LK: Lokiarchaeum sp. GC14_75, PS: Candidatus P. syntrophicum MK-D1, OD: Candidatus Odinarchaeota archaeon LCB_4, HD: Candidatus Heimdallarchaeota archaeon, TO: Candidatus Thorarchaeota archaeon (strain OWC), TA: Candidatus Thorarchaeota archaeon strain AB_25.

while in the CdvBa2 subgroup a conserved proline prohibits this (**Fig 5**). This finding strongly supports the idea of Caspi and Dekker [1] and might point to a solution to the question of membrane tethering in Asgard archaea. Furthermore, the phylogenetic relationship and structural similarity places the Cdv mechanism of Asgard archaea much closer to ESCRT in Eukaryotes than to any other archaeal Cdv-including mechanism.

Based on these potential interactions, we inferred a possible mechanism of cell division in Asgard archaea (**Figs 4D and S1**): First, CdvBa1 proteins bind to the membrane, mediated by the ANCHR domain. Second, both CdvB paralog versions polymerise via the Snf7 domain, giving rise to a filamentous ring-like structure. Third, CdvC disassembles CdvB homologs in the same way as in TACK archaea, constricting the ring. In this final step, our model cannot distinguish whether CdvC first disassembles CdvBa1 or CdvBa2 proteins, because interaction of the proteins can take place either via MIM1-MIT interaction or MIM2-MIT interaction. Furthermore, while in *Cand. P. synthrophicum* the CdvBa1 paralogs seem to possess both MIM1 and MIM2 and the CdvBa2 paralog seems only to possess MIM1, in all other Asgard archaea it is the other way round. There, CdvBa1 proteins only possess MIM1 domains, while CdvBa2 proteins possess correct MIM2 motifs but only fragmented MIM1 domains (**Fig 5**). Thus, differences between the two groups of CdvB in Asgard archaea will have to be investigated experimentally until more high-quality data is available. Also, it might be possible that one of the interactions can lead to binding of CdvC to CdvB without subsequent disassembly.

Finally, as our results suggest mechanistically different Cdv-including mechanisms within the archaeal kingdom, we asked which evolutionary processes might have given rise to these variations and how a simple Cdv machinery in a common ancestor might have looked like. In particular, we wanted to find out if the domain architecture of such an evolutionary early machinery may have been sufficient to give rise to a cell division mechanism.

## An ancestral Cdv-including machinery was probably able to perform constriction, but membrane tethering remains unclear

Based on the structure of the protein phylogenies, the domain architectures and gene cluster analysis (**Figs 2, S2 and S4**), we inferred an evolutionary scenario of domain reorganisation by maximum parsimony (**Fig 6**). Interestingly, this scenario indicates that all CdvB duplication

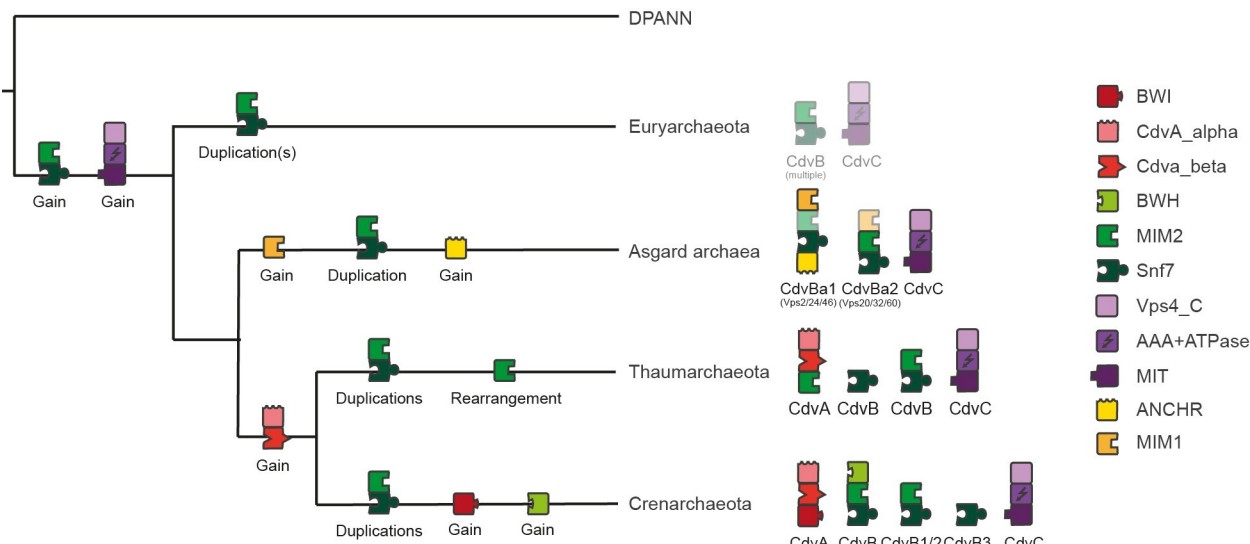

**Fig 6. Potential evolutionary development of Cdv/ESCRT machineries.** Scenario based on phylogenetic trees (**Figs 2** and **S4**) and maximum parsimony. Domains displayed as blocks as in Figs 2–5.

events took place after speciation of the common ancestor into Euryarchaeota, Asgard and TACK, such that in the 'original' machinery there most likely was only one CdvB protein. This protein was probably composed of two domains, Snf7 and MIM2, such that it might have been able to form filaments based on the Snf7 domain. Furthermore, the scenario suggests that the common ancestor also possessed a complete CdvC protein, such that the early CdvB protein might have been able to interact with it via the MIM2 domain. Thus, it is possible that these two proteins together were already able to form a constricting ring by CdvB polymerisation and CdvC-mediated disassembly.

To form a cell division machinery, though, CdvB and CdvC must have been able to attach to the cell membrane. The CdvA protein, which probably fulfils this task in TACK archaea, arose after the speciation of TACK, and the ANCHR domain, which might tether CdvB to the membrane in Asgard archaea, arose only after the speciation of Asgard (**Fig 6**). Thus, it is unclear how the ancient machinery might have fulfilled this task. Interestingly, the region defined as potential ANCHR in Asgard CdvB homologs is also present in TACK CdvB homologs, but it does not form an alpha helix due to a helix-breaking proline. Hence, it is possible that in the evolutionary early CdvB protein this region was functioning as a membrane tether, and that the gain of CdvA made this capability redundant or even obstructive, such that it got lost. If this is true, the ancestral machinery might have been a true cell division machinery.

## Discussion

The mechanism we suggested for Crenarchaeota supports a recent study of Tarrason Risa et al. [6] which investigated the role of CdvB homologs in *S. acidocaldarius*. This organism possesses one protein of the CdvB class, two of the CdvB1/2 class and one of the CdvB3 class. The experiments showed that the CdvB homologs pass four distinct steps: First, the CdvB class proteins assemble at the division site to form a non-constricting ring. Second, CdvB1/2 class proteins are recruited to that ring. Third, CdvB class proteins get disassembled from the division site, leading to a ring consisting of only CdvB1/2 class proteins. Fourth, the CdvB1/2 class ring constricts. Disregarding CdvB3, this is exactly what our model indicates. While Tarrason Risa

et al. described these findings but did not detail the mechanism, our study now provides a plausible scenario of how disassembly of CdvB proteins from the ring can take place and why CdvB class homologs are disassembled before CdvB1/2 class homologs. As the fluorescence microscopy experiment did not yet involve labelled CdvA and CdvC proteins, it would be interesting to see whether our model still holds true in studies investigating the full machinery.

In their paper, Tarrason Risa et al. [6] hypothesised that constriction could be the result of different preferential curvatures of CdvB and CdvB1/2 class proteins. Thus, when CdvB class proteins are removed by CdvC, the ring would shrink to the smaller preferred diameter of CdvB1/2 class proteins. Furthermore, a simulation in this study indicated that to achieve full cell division disassembly of CdvB and CdvB1/2 class homologs is needed in addition to the change in preferred curvature [6]. This explains why both CdvB and CdvB1/2 class homologs possess a MIM2 domain. Furthermore, as we found only one possible interaction for CdvB3 class homologs, namely binding to other CdvB homologs, this need of disassembly suggests that CdvB3 plays a different role. If it cannot interact with CdvC, it cannot be disassembled like CdvB and CdvB1/2. One straightforward idea for the function of CdvB3 might be that it can temporarily stabilise the ring of CdvB and CdvB1/2 homologs while or after CdvC disassembles a component. Simple knockout experiments should provide first insights into the role of CdvB3.

As CdvA was not labelled in their experiment, Tarrason Risa et al. [6] did not comment on how the ring of CdvB1/2 proteins could constrict the membrane without being able to bind to CdvA and thus not being tethered to the membrane. This might be explained by our model. Initially, CdvC mostly disassembles CdvB class proteins due to the potential higher affinity of their MIM2 domain. However, when concentrations change due to disassembly, and more and more CdvC binds to CdvB1/2 class proteins, there should be still some CdvB class proteins left. Thus, even during the final constriction stage, there should always be some remaining membrane anchors, as CdvB class proteins are never entirely removed. It will be very interesting to test the idea of different MIM2-MIT affinities.

There is, however, one result from another study which cannot be explained by our model. Based on data taken with a newly developed microscope, Pulschen et al. [5] suggested that the two different class CdvB1/2 proteins of *S. acidocaldarius*, called CdvB1 and CdvB2, may play different roles. While they colocalised in their experiments as expected, cells lacking CdvB1 did occasionally fail cell division, whilst cells lacking CdvB2 showed a wrongly positioned division site. As we classified them as two paralogs belonging to the same group, we cannot explain this behaviour. However, we did not investigate this in detail, as it is a specific modification in one particular subgroup of Crenarchaeota. Here, we tried to infer common shared principles, which might then be slightly adapted individually by single organisms. Furthermore, none of the proteins were essential, as would be expected from our model. Unfortunately, they did not test cells lacking both CdvB1 and CdvB2, which would be very interesting for validating the theory of membrane constriction due to different preferred curvatures. Such double-knockout experiments would greatly help understanding the mechanism of constriction.

To conclude, in this paper we stated two possible scenarios for Cdv-including cell division in Thaumarchaeota. While we strongly favour one of them, only experimental work can reveal which one comes closer to reality. As in one scenario CdvB homologs are involved, while in the other they are not, the most straightforward way to answer this question might be to knock out all CdvB homologs and see whether cell division is still possible.

While our proposed models of cell division in TACK archaea provide a theoretical basis that should inspire experimentalists to new studies, the results also show that findings in TACK archaea are only poorly transferable to the eukaryotic ESCRT system. However, we demonstrated the high potential of research in Asgard archaea, whose Cdv machinery we

found to be closely related to ESCRT. Based on our results we recommend two *in vitro* experiments that might be possible to conduct despite the hard task of Asgard cultivation. First, as we suggested that CdvBa1 proteins can bind to the membrane via their ANCHR domain, whilst CdvBa2 proteins cannot, this should be tested *in vitro* to verify the central idea of our model. Second, as we could not quantify differences between CdvBa1 and CdvBa2 paralogs in their affinity to CdvC, experiments providing insight into this would greatly help to evolve the model further. In addition, a recent study by Hatano et al. [13] further suggests that at least some Asgard archaea also possess functional homologs of the additional proteins involved in the eukaryotic ESCRT machinery, ESCRT-I and ESCRT-II, and that they might be involved in ubiquitin-directed recruitment of CdvB (ESCRT-III). While we did not investigate these additional components here, as our main goal was to find similarities between all archaea that could be traced back to a very simple division machinery, this study once more emphasises the strong connection between Asgard and the eukaryotic machineries.

The close relationship between Cdv proteins in Asgard archaea and Eukaryotes further allows an interesting hypothesis about membrane binding in the ancient Cdv machinery in the common ancestor of all Cdv machineries. The potential ANCHR domain in Asgard archaea suggests that some CdvB homologs in these organisms can directly bind membranes, just as some ESCRT-III proteins in Eukaryotes. On the other hand, recent analyses of the membrane composition of Asgard archaea suggest that their lipid composition is comparable to other archaea, strongly differing from Bacteria and Eukaryotes. The important question is now whether the analogy of ANCHR motifs for membrane binding in Eukaryotes and Asgard is still valid, despite the differing lipid composition. If so, a direct membrane tethering of CdvB proteins may have occurred in the common ancestor, whose membrane probably consisted of similar lipids as archaea today. If not, it would be interesting to explore why Asgard archaea possess ANCHR domains and what task apart from membrane-binding they could serve. Some CdvB homologs in TACK archaea possess a region similar to and aligning with the region identified as ANCHR domain in Asgard homologs, but it cannot form an alpha helix due to a helix-breaking proline. As this region is present in diverged organisms, it is likely that it already existed in the common ancestor. The question is now whether it then was able to form an alpha helix to bind the membrane. If so, then the ancient machinery might have been able to function as a true cell division machinery, consisting of only two proteins.

Beyond cell biology and evolution, the reconstruction of this very simple ancient division machinery should be of great interest for the synthetic biology community. Bottom-up synthetic biologists try to build life from scratch, and especially to build basic functional building blocks *in vitro*. One such basic functional building block would be a simple system able to divide a lipid vesicle, mimicking true cell division. A synthetic system reconstituting the ancient Cdv machinery inferred by this study should be a very promising candidate for this yet unaccomplished endeavour. Especially, it will be very interesting to fabricate synthetic ESCRT-III/CdvB proteins consisting of one Snf7 and one MIM2 domain and to combine them with Vps4 or CdvC proteins *in vitro*, possibly even inserting them into giant unilamellar vesicles. If successful, this would depict an extremely simple system of cell division, that might be easy to study and allow interesting insights into the origin of cell division and life.

## Materials and methods

### Searching Cdv homologs

Archaeal proteins classified in the eggNOG [30] database as belonging to the Cdv system were used as starting points together with the proteins investigated by Makarova et al. [2] in a previous study. These included the entries ENOG4111F6A (CdvA), COG5491 (all CdvBs) and

COG0464 (CdvC), corresponding to entries arCOG04054 (interestingly CdvA and B3), arCOG00452 (cdvB1/2), arCOG00453 (CdvB) and arCOG01307 (CdvC) of the archaea-only database arCOG [31]. Using the resulting set of proteins as input, a Python script was designed to run PSI-Blast [32] against the NCBI protein [33] database (v2020-11-11). Default parameters were used except that the scoring matrix was set to BLOSUM45 [34] due to the distant relationships. Then, Python scripts were written to eliminate all hits from organisms with unclear phylogeny, organisms with fragmented genome data, results of marine sediment probes and duplicates from different strains of the same species. Only in Asgard archaea incomplete genomes were accepted and multiple strains included, because all available data is incomplete except for *Cand. P. syntrophicum*. Hits to CdvC homologs were then further analysed because the AAA+ ATPase region is widely spread amongst many different proteins and accounts for most of the sequence of CdvC proteins. Thus, many false positive hits that are no true CdvC homologs could be expected. Only sequences that possessed at least a VPS4_C or a MIT region were further selected (exception: *F. acidarmanus*, because it is part of the arCOG01307 entry and had a very good PSI-Blast E-Score of 2e-48). Finally, proteins in close genomic neighbourhood of all finally selected proteins were checked whether they may depict overseen homologs. Eventually, 37 organisms remained and were selected for further analysis. The set of proteins can be found in **S2 Table** (which also includes FtsZ proteins).

## Fragmenting protein sequences into domains

To fragment protein sequences into domains we used two different approaches. On one hand, we used available software tools to scan the proteins' sequences for domains that are described in public databases. For scanning against the Pfam [35] and InterPro [36] databases we used the InterProScan [37] software, while for scanning against the CDD [38] database we used a custom Python script connecting to the online API of the database. The resulting data files were then processed to summarise hits that represented the same domains or were sub-parts of a larger domain. This led to the identification of the domains CdvA_alpha, BWH, Snf7, MIT, AAA+ ATPase and Vps4_C. On the other hand, we had to check for domains and short conserved motifs which are not defined in the three mentioned databases. For this, we generated multiple sequence alignments and predicted the secondary structure of the proteins. We then inspected the results manually for regions with high sequence conservation and consistent predicted secondary structure, which led to the identification of a beta-sheet rich region in TACK archaeota CdvA proteins, a single conserved beta-sheet in Crenarchaeota CdvA proteins, and the conserved alpha helices in Asgard CdvB proteins. As the beta-sheet rich region in CdvA proteins was previously described as a functional region [1,16,18,19], we labelled it as a unique domain, named CdvA_beta. Similarly, the conserved single beta-sheet in Crenarchaeota CdvA proteins was previously described as functional region for interaction with the BWH domain in CdvB [18], so we classified this as domain, named BWH interaction site (BWI). Furthermore, inspired by Caspi and Dekker [1] and Lu et al. [22], we then compared the conserved regions in Asgard archaea to the N-terminal ANCHR domain [24] and the C-terminal MIM1 domain [23] occurring in some ESCRT-III proteins (**Fig 5**). This led to the identification of the ANCHR and MIM1 domains. Finally, based on the work of Kojima et al. [29] and Samson et al. [18], a python script was generated to search for the MIM2 motif by regular expressions. Three different kinds of MIM2 were defined: MIM2_Core = φPxφP, MIM2_total = MIM2_Core + xxPφP and MIM2_Sulf = xφxxφφPx + MIM2_Core, where φ represents hydrophobic amino acids [AILMFVPGW], x represents charged amino acids [RKDE] and P is proline. In the two Thaumarchaeota *Nitrososphaera viennensis* and *Candidatus Nitrosphaera gargensis* we classified a proline-rich region as 'putative MIM2', although not

perfectly fitting to the regular expressions. This is justified because they aligned to the proline rich regions of the Thaumarchaeota possessing a full MIM2 motif, they did not show other proline-rich regions and the structural limitations of MIT-MIM2 interaction are unknown despite being based on proline. Taken together, these analyses led to the eleven basic building blocks that are utilized in the Cdv system. **S1 Table** provides an overview of the exact location of the domains on the proteins.

## Multiple sequence alignment and secondary structure prediction

The Multiple Alignment tool MAFFT [39] was used for sequence alignment. For proteins belonging to different super-phyla (different background colours in Fig 1) the substitution matrix BLOSUM45 [34] was used, for alignment of proteins of the same super-phyla BLOSUM62 [34]. The exact command-line instruction was

mafft—bl 45 (or 62)—localpair—maxiterate 1000—reorder input.fasta > output.fasta.

Secondary structure prediction based on multiple sequence alignment was executed with JPred [40] using default parameters.

## Phylogenetic analysis

Based on the multiple sequence alignments, Bayesian phylogenetics was used to generate phylogenetic trees. The software Mr. Bayes [41] was run with a chain length of 1,000,000, a subsample frequency of 1,000, a burn-in length of 100,000, gamma rate variation with 4 categories and a Poisson rate matrix. The resulting effective sample size was 213 resp. 181.

## Mechanism inference

Each protein was abstracted as a combination of domains (**Figs 1, 4 and S1**). Based on literature, each domain was assigned a specific function (**Fig 4**). Then, networks of possible interactions were built, based on potential interactions derived from the literature (**Figs 3 and S1**). To infer a potential mechanism based on these interaction networks, one starting protein was selected for each mechanism. In TACK archaea, CdvA homologs were selected, because fluorescence microscopy studies indicate that they enrich first at the division site (1,6,18). Also, they are the only Cdv proteins able to bind the membrane in TACK archaea, and they possess a CdvA_beta domain, which we did not assume to interact with other Cdv proteins. Thus, this domain has the potential to be involved in the positioning of CdvA. Neither CdvB homologs nor CdvC did show such "unused" domains in TACK archaea, so if instead these proteins would enrich first at the division site, we could not explain this by domain architecture. In Asgard archaea, we selected CdvBa1 as starting protein, because it is potentially able to bind membranes. However, it is of course possible that instead CdvBa2 enriches first. This would change the order of appearance, but the overall assumed mechanism of polymerisation, membrane binding and CdvC mediated depolymerisation would still hold.

The subsequent process of mechanism inference is visualised in **S1 Fig**. After selecting a starting protein, we analysed which interactions were possible for this protein. If only one interaction was possible, we assumed this to occur as a next step. If more than one interaction was possible, we tried to infer which interaction might have a higher affinity, and thus would occur more quickly. If this was not possible, we inferred different scenarios. These steps were then executed recursively. The resulting mechanisms were then summarized and simplified to the mechanisms displayed in **Fig 1**.

### Code availability

The used data and python scripts are available at a public GitHub repository (https://github.com/BelaFrohn/ArchaealESCRTDomains).

## Limitations of the study

The study is purely computational with the goal to generate hypotheses that can be tested experimentally. Thus, the described mechanistical models are suggestions and must be validated experimentally. Furthermore, we based our analysis purely on domain analysis, so any functional changes that result from small sequence or structural alterations that are not reflected in domain architecture are not incorporated in our models. Such small alterations most likely cause small functional differences between highly similar homologs, hence between closely related organisms. Consequentially, our analysis cannot draw detailed conclusions about single particular organisms. We were interested in general principles, whose implementation in distinct organisms might differ slightly. To infer explicit models for single organisms, more detailed structural modelling of the involved proteins is necessary.

## Supporting information

**S1 Fig. Interactions underlying the suggested mechanisms.** In Crenarchaeota, the potential polymerisation capabilities of CdvA and CdvC are not displayed as they do not help explain the suggested mechanism. In Asgard archaea, the CdvC polymerisation capability is likewise not displayed. Background colours indicate phylogenetic groups. Red: Asgard archaea, dark green: Thaumarchaeota, pale green: Crenarchaeota.
(TIF)

**S2 Fig. Cluster analysis of Cdv genes in archaea.** Organisms are grouped by phylogeny. Genes encoding CdvA, CdvB and CdvC are in direct neighbourhood in Crenarchaeota. In Thaumarchaeota, in three out of four organisms one CdvB homolog gene is in close neighbourhood to the CdvC gene. CdvA homologs are not. In Euryarchaeota, if there are genes encoding both CdvB and CdvC homologs, they are either in direct or in relatively close neighbourhood. In Asgard archaea, except in *Cand. P. syntrophicum* only fragmentary genomes are available. In three out of six Asgard archaea genomes CdvC and CdvB homolog genes are in close neighbourhood. This pattern across all archaeal phyla indicates that indeed CdvC and one CdvB homolog existed in the last common ancestor of TACK, Asgard and Euryarchaeota.
(TIF)

**S3 Fig. Multiple Alignments of the MIM2 domain in CdvB and CdvB1/2 proteins.** In CdvB1/2 proteins the MIM2 domain contains consistently fewer prolines, indicating a weaker affinity to the MIT domain than the MIM2 of CdvB proteins. Abbreviations: AP: Aeropyrum pernix, AS: Acidilobus saccharovorans, PO: Pyrodictium occultum, HB: Hyperthermus butylicus, AH: Acidianus hospitalis, AM: Acidianus manzaensis, AS: Acidianus sulfidivorans, SI: Sulfolobus islandicus, SS: Saccharolobus solfataricus, SA: Sulfolobus acidocaldarius, ST: Sulfurisphaera tokodaii, MS: Metallosphaera sedula, FF: Fervidicoccus fontis, DA: Desulfurococcus amylolyticus, SM: Staphylothermus marinus, CL: Caldisphaeralagunensis, IH: Ignicoccus hospitalis.
(TIF)

**S4 Fig. Phylogenetic Tree of CdvC proteins.** The overall structure shows the same phylogenetic relations as the CdvB tree (**Fig 2**): Asgard archaea are located between Euryarchaeota and Crenarchaeota, while Thaumarchaeota are most distant from Asgard archaea. However, the

Euryarchaeota group of Thermoplasma is not located near all other Euryarchaeota but between Asgard archaea and TACK archaea. This might be an artifact because the proteins in Thermoplasma mutated massively and maybe gained an entirely new function (the branch is the longest in the whole phylogeny), or it might be caused by horizontal gene transfer. Background colours indicate phylogenetic groups. Blue: Euryarchaeota, red: Asgard archaea, dark green: Thaumarchaeota, pale green: Crenarchaeota. Numbers indicate posterior probabilities. (TIF)

**S1 Table. Domain locations of the investigated proteins.**
(XLSX)

**S2 Table. Protein and gene data of the investigated proteins.**
(CSV)

## Acknowledgments

We thank Dina Grohmann (University of Regensburg) and Henri Franquelim for helpful feedback on the paper. B.F. is supported by the Graduate School of Quantitative Biosciences Munich (QBM).

## Author Contributions

**Conceptualization:** Béla P. Frohn, Jürgen Cox, Petra Schwille.

**Formal analysis:** Béla P. Frohn, Tobias Härtel.

**Investigation:** Jürgen Cox, Petra Schwille.

**Software:** Béla P. Frohn.

**Supervision:** Jürgen Cox, Petra Schwille.

**Visualization:** Béla P. Frohn.

**Writing – original draft:** Béla P. Frohn.

**Writing – review & editing:** Béla P. Frohn, Jürgen Cox, Petra Schwille.

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
