## [Decision Letter · Decision Letter 0]

5 Jan 2022

PONE-D-21-29711

Tracing back variations in archaeal cell division to protein domain architectures

PLOS ONE

Dear Dr. Cox,

Thank you for submitting your manuscript to PLOS ONE. After careful consideration, we feel that it has merit but does not fully meet PLOS ONE’s publication criteria as it currently stands. Therefore, we invite you to submit a revised version of the manuscript that addresses the points raised during the review process.

We look forward to receiving your revised manuscript.

Kind regards,

Ernesto Perez-Rueda

Academic Editor

PLOS ONE

2. Please note that in order to use the direct billing option the corresponding author must be affiliated with the chosen institute. Please either amend your manuscript to change the affiliation or corresponding author, or email us at plosone@plos.org with a request to remove this option.

Reviewers' comments:

Reviewer's Responses to Questions

**Comments to the Author**

1. Is the manuscript technically sound, and do the data support the conclusions?

Reviewer #1: Partly

Reviewer #2: Yes

2. Has the statistical analysis been performed appropriately and rigorously? 

Reviewer #1: Yes

Reviewer #2: N/A

3. Have the authors made all data underlying the findings in their manuscript fully available?

Reviewer #1: No

Reviewer #2: Yes

4. Is the manuscript presented in an intelligible fashion and written in standard English?

Reviewer #1: Yes

Reviewer #2: Yes

5. Review Comments to the Author

Reviewer #1: The authors discuss an important topic in the article, such as the cell division machinery in Archaea, with a specific focus on the component of the Endosomal Sorting Complex Required for Transport. They employed a bioinformatics technique to develop their investigation at the level of domains and protein motifs in the homologous sequences of a group of Archaea.

The study has a good development; however, several questions emerge from the text that must be addressed before the study is accepted for publishing.

The greatest problem emerges in sentences like:

142

230-234

318-321

where, despite the fact that it is entirely bioinformatic study, additional analyses are required to allow consolidation of their considerations

Some other considerations are listed below.

In 131: They selected 37 organisms for their analysis. But in Figure 1 and Suppl. Table 1. The set of organisms that they present corresponds to a set of 51 organisms.

In 133: CDD needs to be defined the first time it is used.

In 135: Figure 1B does not contain the eleven domains that they describe in the text.

In general, figure captions are very extensive and tend to repeat information from the main text. As a result, they should be reorganized in general.

In this context, Figure 1 includes too much information forcing the transfer of a component to Figure 3. It is preferable to create one more figure that makes the information more clear; this will help to better understand the phylogenetic information. Finally, the legend of the figure must be written in a sequential manner and must avoid discussing portion B) at the conclusion.

They indicate that CDv genes are identified in just a few species in the case of Euryarchaeota. It would be interesting to study if these remaining genes have been subjected to relaxed selection pressures. To back up the decision expressed in lines 200-202.

In 238, change T to t.

If the interactions described are previously known, the section 230-242 should be appropriately cited, and if they are the results of the study, further proof of these interactions should be provided.

Regarding figure 2, it is unclear if the phylogeny was constructed using the 37 organisms or the 51 in figure 1, and how the organisms in the Eukaryotic clades were chosen. Check the Eukaryotic typeface in figure 2.

Figure 3 shows the domain table with information that is duplicated from Figure 1B.

Although the article does not focus on other types of machinery, it would be preferable for the authors to address more fully what happens with organisms that do not conserve CDv genes.

Finally, all data, including scripts, should be included as supplemental material or deposited in a freely accessible repository.

Reviewer #2: Cox et al. present a detailed in silico characterization and bibliographic revision of the proteins of archaeal ESCRT machineries (i.e. CdvABC) and compares them with their better described Eukaryotic counterparts to infer a series of models for the mechanisms by which these might be involved in cell division withing the Euryarchaeota, Asgard and TACK super-phyla. Also, by finding similarities between all archaea that could be traced back to a very simple division machinery, this study adds yet another piece of evidence for the close relationship between Asgard and the eukaryotic machineries.

Overall the rational of the authors is sound and it was a very pleasant read for someone the is not a specialist in Archaeal cell division mechanisms. This mauscript builds upon the work of Tarrason Risaet et al. (2020) and the models presented here seem like a step in the correct direction regarding the mechanistic characterization of the CdvABC proteins. However, I would like to ask for the authors to create and share with us a robust and reproducible (bioinformatic) algorithm or tool (script) to identify Archaeal CdvABC proteins. While reading their methods it sometimes feels like they are cherry-picking some protein sequences and discarding others. That is only the feeling I get when reading this version of the manuscript and I would greatly appreciate something like a validated HMM that could be offered to the community for these Archaeal proteins.

General comments

Are the assignments of the protein domains described in the article similar to what tools like InterPro scan reveal as output using HMMs? and, if no Pfam model were available. would it be possible to obtain the same results when creating HMMs using your set of known domains instead of regular expressions?

I would ultimately like to know if your approach is as sensitive and/or robust as the extensive use of HMMs to determine a proteomic domain or motif.

Resolution of the Figures I reviewed was very low and I hope that this is due to the Plos review platform and not the authors’ design. Figures 1 and 3 are a vital component of this article and should have exceptional resolution to offer the most needed aid the reader requires to understand the different variations of the CdvABC secondary structures and their proposed mechanisms.

Please note that major parts of the “Figure legends” should be moved to the “Results” or “Discussion” sections of the main text as descriptions of the outcome of your analysis, and not as the caption describing the figure to the reader.

Specific comments

In line no. 14 “... This spread across super-phyla suggests that…” Archaea and Eukaryotes have the taxonomic rank of “Domains” of life, not “Super-phyla”, specify that you assessed “super-phyla within these Domains”.

In line no. 68 “... After the discovery of CdvA, -B and -C in S. acidocaldarius, …” Better refer to these three proteins as “CdvABC” here and below.

In line no. 82 “... concept as one highly refined version of the Asgard Cdv system. This relationship emphasizes that it is necessary to investigate archaeal Cdv machineries in addition to eukaryotic ESCRT machineries to draw conclusions about the ancestral machinery.”

This phrase would read better if written something like:

In line no. 101 “... On the one hand, experimental studies with archaeal Cdv machineries are technically highly challenging, due to the extreme conditions archaeal organisms thrive under. On the other hand,...” These phrases would read better if you started them with “First” and Second” instead of “on one hand and on the other”.

In line no. 150 “... Thus, this simple analysis already yielded two interesting results: First, four domains are shared between all organisms…” Please constrain the extent of your findings to “all genomes included in this analysis” instead of “all organisms”.

In line no. 161 I recommend a more universal color name instead of “Khaki”, such as “yellow” or “pale yellow”.

In Figure 1A legend please move to the Results section descriptions of the outcome of your analysis, such as those from line no. 163 “Differing domain patterns matching the phylogenetic groups are clearly visible…” to line no. 177. There are more phrases further below that can also be moved to results instead of the figure caption.

248 “Thus, these previously unexplained experimental differences fit nicely to our analysis of potential PPIs.” I suggest changing “fit nicely” with “are further supported by” or something similar.

Please rephrase line no. 249 “potential PPIs. Furthermore, it was observed that in Crenarchaeota it is indeed the CdvB homolog with the BWH domain that enriches first, followed by the other CdvB homologs (5).”

In line no. 261 “Reassuringly, the differences in domain architecture fitted perfectly to the phylogenetic clusters (Figure 2).” change to something like “... were congruent with their phylogeny, as seen by their placement in different branches within the CdvB tree”

Please tone down the following phrase (that should be in Results section and not as part of a Figure legend) to clarify that this is an hypothetical explanation derived from your data (e.g. “Our data supports that…”): in line no. 277 “This means that all interactions between different CdvB homologs developed independently and are therefore not comparable.”

Text from line 276 to 286 should be incorporated to the “Results” or “Discussion” sections of the main text as descriptions of the outcome of your analysis

Please change line no. 380 “Our analysis of multiple Asgard archaea genomes and especially the trustworthy Cand. P. synthrophicum genome…” with something like “Our analysis of multiple Asgard archaea genomes and especially that of its only cultivated member Cand. P. synthrophicum…”

Text from line 399 to 406 should be incorporated to the “Results” or “Discussion” sections of the main text as descriptions of the outcome of your analysis

Text from line 461 to 467 should be incorporated to the “Results” or “Discussion” sections of the main text as descriptions of the outcome of your analysis

Please change in line no. 481 “The mechanism we suggested for Crenarchaeota fits nicely to a recent study of Tarrason Risa et al. (6)” with “... further supports the recent study of…”

In line no. 594 “... to run PSI-Blast (27) against the NCBI protein (28) database.” Please specify which NCBI database version you used (e.g. v2021-12-20)

In line no. 610 “Decomposing proteins into domains” Please consider changing this to “Fragmented analysis of protein domains” or something like that (here and along all the methods section below)

6. PLOS authors have the option to publish the peer review history of their article (what does this mean?). If published, this will include your full peer review and any attached files.

Reviewer #1: No

Reviewer #2: **Yes: **Alvaro M. Plominsky

---

## [Author Response · Author response to Decision Letter 0]

15 Feb 2022

Our replies are contained in the uploaded document Response to Reviewers.pdf.

---

## [Decision Letter · Decision Letter 1]

21 Mar 2022

Tracing back variations in archaeal ESCRT-based cell division to protein domain architectures

PONE-D-21-29711R1

Dear Dr. Cox,

We’re pleased to inform you that your manuscript has been judged scientifically suitable for publication and will be formally accepted for publication once it meets all outstanding technical requirements.

Kind regards,

Ernesto Perez-Rueda

Academic Editor

PLOS ONE

Additional Editor Comments (optional):

Reviewers' comments:

Reviewer's Responses to Questions

**Comments to the Author**

1. If the authors have adequately addressed your comments raised in a previous round of review and you feel that this manuscript is now acceptable for publication, you may indicate that here to bypass the “Comments to the Author” section, enter your conflict of interest statement in the “Confidential to Editor” section, and submit your "Accept" recommendation.

Reviewer #1: All comments have been addressed

Reviewer #2: All comments have been addressed

2. Is the manuscript technically sound, and do the data support the conclusions?

Reviewer #1: Yes

Reviewer #2: Yes

3. Has the statistical analysis been performed appropriately and rigorously? 

Reviewer #1: Yes

Reviewer #2: N/A

4. Have the authors made all data underlying the findings in their manuscript fully available?

Reviewer #1: Yes

Reviewer #2: Yes

5. Is the manuscript presented in an intelligible fashion and written in standard English?

Reviewer #1: Yes

Reviewer #2: Yes

6. Review Comments to the Author

Reviewer #1: (No Response)

Reviewer #2: General comments

I liked the current experimental design and bioinformatic analysis supporting the nice findings the authors made. The logic was sound and the authors propose a guide as well as interesting challenges for experimental biologists to test.

I noticed that the quality of the manuscript increased notably after line 190. Please revise the writing for the first 190 lines again as there are better ways to present your ideas and the results of your work. I gave some recommendations but this is not an exhaustive list.

The resolution of the Figures I reviewed was very low, but I downloaded Figures from the github associated with this article and all figures were clear and high quality. Please make sure that quality is translated into the final article version. All main Figures are a vital component of this article and should have exceptionally high resolution to offer the most needed aid the reader requires to understand the presence/absence of Cdv proteins in the various taxa and the different variations of the CdvABC domains and their proposed interactions.

Specific comments

In line no. 57 “... because a comparative approach investigating similarities of machineries derived from the same ancestor is the more interesting the more different descendants are included.” I suggest changing to “... would offer more significant insight when including a higher diversity of descendants in the analysis.”

In line no 67 “... Thus, there are both similarities and differences in the composition of the machineries, fitting to the hypothesis that they derived from a common ancestral machinery but evolved differently.” I suggest changing to “... supporting the hypothesis that they derived from a common ancestral machinery…”

In line no 71 “... Interestingly, for which of the three proteins homologs were found was closely related to the phylogeny of Archaea.” I suggest removing that sentence.

In line no 82 “... phyla once more indicate that there was once an ancient Cdv/ESCRT machinery that has been adapted differently.” I suggest changing to “... evolved differently.”

In line no 85 “... This idea emphasizes that if one wants to draw conclusions about an ancestral ESCRT machinery, it is essential to investigate archaeal Cdv machineries in addition to eukaryotic ESCRT machineries.” I suggest changing to “... This idea emphasizes the relevance of studying archaeal Cdv machineries to increase our understanding of ancestral versions of the eukaryotic ESCRT machineries.”

In line no 114-119 “... This is inspired by the idea that the mechanism of a protein machinery is a result of the interactions between the proteins, and that these interactions are located at specific regions of the proteins’ amino acid sequences. We show that CdvABC homologs in Cren- and Thaumarchaeota differ in their domain compositions and that the different interactions of the proteins caused by these different compositions have the potential to explain the variations found in experiments.” I suggest changing to “... and that these interactions occur at specific regions of their amino acid sequences… and that the distinctive interactions of the proteins caused by these different compositions might explain the previously observed experimental variations.”

In line no 134-139 (sentence is too long) I suggest changing it to something like: “... From the resulting list of organisms containing at least one CdvABC homolog we selected 37 organisms based on the high quality of their genomic information. The secondary structure of the Cdv protein architecture from these selected genomes was analyzed by processing them through Pfam, InterPro, CDD, and utilizing previously reported regular expressions to identify domains of interest. Hereafter we will call all regions in the amino acid sequences of proteins that are expected to execute a distinct function ‘domains’, although strictly speaking some are only motifs.”

In line no 155-156 “... First, four domains are shared between all organisms included in this study, which suggests that the common ancestor of Cdv-including systems might have utilized these domains.” I suggest changing to “..., which suggests that the Cdv present in their common ancestor likely presented these domains.”

In line no 181-183 It’s better not to speculate on what evolutionary event was responsible for mutating the Euryarchaeota cdv genes if you are not going to test these hypotheses. Just state they “... mutated beyond recognition.”

In line no 184-185 “Furthermore, no domains allowing interaction between proteins could be identified, making a system similar to Cdv-including mechanisms in other archaea unlikely.” I suggest changing to something like “... Furthermore, no domains allowing interaction between proteins could be identified. Thus, suggesting that Euryarchaeota likely have a different mechanism for cell division.”

Please make the “Crenarchaeota” label under its corresponding set of interaction models in Fig. 2 more visible (darker) because it was barely notable in the figure set I received to review.

In line no 325 “...Which of these options is utilised first cannot be inferred from our analyses…” change to “utilized”

In line no 367 “...Next, CdvC binds to CdvA, utilising the MIM2-MIT interaction.” change to “utilizing”.

In line no 605 “...(exception: F. acidarmanus, because it is part of the arCOG01307 entry and had a very good PSI-Blast score).” Please specify the score instead of saying it is “very good”.

7. PLOS authors have the option to publish the peer review history of their article (what does this mean?). If published, this will include your full peer review and any attached files.

Reviewer #1: **Yes: **Edgardo Galán-Vásquez

Reviewer #2: **Yes: **Alvaro M. Plominsky

---

## [Editor Report · Acceptance letter]

23 Mar 2022

PONE-D-21-29711R1 

Tracing back variations in archaeal ESCRT-based cell division to protein domain architectures 

Dear Dr. Cox:

I'm pleased to inform you that your manuscript has been deemed suitable for publication in PLOS ONE. Congratulations! Your manuscript is now with our production department. 

Kind regards, 

on behalf of

Dr. Ernesto Perez-Rueda 

Academic Editor

PLOS ONE